# Antibody–Drug Conjugates Powered by Deruxtecan: Innovations and Challenges in Oncology

**DOI:** 10.3390/ijms26136523

**Published:** 2025-07-07

**Authors:** Jung Yoon Jang, Donghwan Kim, Na Kyeong Lee, Eunok Im, Nam Deuk Kim

**Affiliations:** 1Department of Pharmacy, College of Pharmacy, Research Institute for Drug Development, Pusan National University, Busan 46241, Republic of Korea; jungyoon486@pusan.ac.kr (J.Y.J.); nklee@pusan.ac.kr (N.K.L.); 2Functional Food Materials Research Group, Korea Food Research Institute, Wanju-gun 55365, Jeollabuk-do, Republic of Korea; kimd@kfri.re.kr; 3Department of Pharmacy, Eson Geriatric Hospital, Ulsan 44955, Republic of Korea

**Keywords:** antibody–drug conjugates, deruxtecan, topoisomerase inhibitor, anticancer

## Abstract

Antibody–drug conjugates (ADCs) have revolutionized precision oncology by enabling targeted drug delivery with improved therapeutic indices. Among these, deruxtecan (DXd)-based ADCs have demonstrated remarkable efficacy across a range of cancers, particularly in tumors expressing human epidermal growth factor receptor 2 (HER2), human epidermal growth factor receptor 3 (HER3), and trophoblast cell surface antigen 2 (TROP2), including breast, lung, gastric, and other solid tumors. DXd, a potent topoisomerase I inhibitor, enhances the cytotoxic potential of ADCs through a cleavable and stable linker and a high drug-to-antibody ratio that ensures optimal drug release. The clinical success of trastuzumab DXd (Enhertu^®^) and datopotamab DXd (Datroway^®^), along with the ongoing development of patritumab DXd, has expanded the therapeutic potential of ADCs. However, challenges remain, including toxicity, resistance, and manufacturing scalability. This review discusses the mechanisms of action, clinical progress, and challenges of DXd-based ADCs, highlighting their transformative role in modern oncology and exploring future directions to optimize their efficacy and accessibility.

## 1. Introduction

The advent of precision medicine has significantly transformed cancer treatment, with antibody–drug conjugates (ADCs) playing a pivotal role in this evolution. These biopharmaceuticals combine the antigen specificity of monoclonal antibodies with the potent cytotoxicity of chemotherapeutic agents, thereby enhancing therapeutic efficacy while minimizing systemic toxicity [1]. Advances in antibody engineering, linker chemistry, and payload optimization have propelled the development of ADCs, resulting in the approval of multiple agents across a variety of malignancies [2]. Among the most recent breakthroughs in ADC technology, deruxtecan (DXd)-based ADCs have demonstrated exceptional efficacy in multiple cancer types, particularly those expressing human epidermal growth factor receptor 2 (HER2), human epidermal growth factor receptor 3 (HER3), and trophoblast cell surface antigen 2 (TROP2) [3,4,5]. DXd, a potent topoisomerase I inhibitor, disrupts DNA replication, thereby enhancing the cytotoxic effects of ADCs. This mechanism not only improves targeted drug delivery but also maximizes the overall therapeutic potential of DXd-based ADCs [6,7].

The role of topoisomerase inhibitors in oncology is well established, particularly in combination chemotherapy for colorectal and breast cancers, where their ability to enhance DNA-damage-induced cytotoxicity has been extensively investigated [8]. Notably, the mechanism of action of DXd shares similarities with other topoisomerase inhibitors, such as etoposide, a topoisomerase II inhibitor widely used in lung cancer treatment. Etoposide exerts its effects by inducing DNA strand breaks, which lead to apoptosis and cell cycle arrest [9]. Additionally, emerging topoisomerase inhibitors, such as MHY440, which induces cell cycle arrest and apoptosis via a reactive oxygen species (ROS)-dependent DNA damage signaling pathway, highlight the continued expansion of topoisomerase-targeted therapeutics [10]. Unlike earlier-generation ADCs, DXd-based ADCs feature a high drug-to-antibody ratio (DAR) and a stable linker system. Furthermore, their optimized bystander effect enables the drug to exert cytotoxic activity beyond antigen-expressing tumor cells, effectively targeting neighboring tumor cells as well [11].

As of June 2025, two DXd-based ADCs have received regulatory approval: Enhertu^®^ and Datroway^®^ [12,13,14,15]. Trastuzumab deruxtecan (T-DXd, DS-8201a, Enhertu^®^) is the most extensively studied agent in this class, designed to selectively target HER2-expressing tumors [16]. It was first approved by the U.S. Food and Drug Administration (FDA) in December 2019 for the treatment of HER2-positive breast cancer, with subsequent approval by the European Medicines Agency (EMA) for the same indication in January 2021 [12,13]. Clinical trials have demonstrated its superior efficacy, particularly in patients who have progressed following prior HER2-targeted therapies [17]. In addition to breast cancer, T-DXd is being actively investigated in other HER2-expressing malignancies, including gastric, lung, and colorectal cancers, thereby expanding its therapeutic potential [16].

Datroway^®^ (datopotamab deruxtecan, Dato-DXd, DS-1062a) has also received regulatory approval from both the FDA and the EMA for the treatment of unresectable or metastatic hormone receptor (HR)-positive, HER2-negative breast cancer in patients previously treated with endocrine therapy and chemotherapy [14,15]. The FDA granted approval on 17 January 2025, based on results from the Phase III TROPION-Breast01 trial, which showed a 37% reduction in the risk of disease progression or death compared to chemotherapy [18]. Following a positive recommendation by the Committee for Medicinal Products for Human Use (CHMP) on 30 January 2025, the European Commission granted marketing authorization in the EU on 8 April 2025 [19]. Clinical studies have shown promising antitumor activity, particularly in TROP2-overexpressing cancers such as triple-negative breast cancer (TNBC) and non-small cell lung cancer (NSCLC) [20,21].

In addition to these approved DXd-based ADCs, new candidates are expanding the therapeutic landscape. Patritumab DXd (P-DXd, U3-1402, HER3-DXd) is engineered to target HER3, a receptor increasingly recognized for its oncogenic role [22]. Early-phase clinical studies suggest promising efficacy, particularly in NSCLC and breast cancer, where HER3 overexpression is frequently observed [4,23]. Similarly, ifinatamab DXd (I-DXd, DS-7300a) is a B7-H3-targeting ADC that has demonstrated promising preclinical and early clinical efficacy, especially in solid tumors with high B7-H3 expression such as lung and prostate cancers [24]. DS-6157a is a G protein-coupled receptor 20 (GPR20)-directed ADC, and it was previously under investigation for the treatment of gastrointestinal stromal tumors (GISTs). However, the clinical development was discontinued after Phase I due to limited efficacy, despite acceptable safety outcomes [25]. This case underscores the variability in therapeutic success across DXd-based ADCs and highlights the need for careful target selection and disease-specific validation. Another emerging DXd-based ADC, raludotatug DXd (R-DXd, DS-6000a), targets cadherin-6 (CDH6) and is being evaluated in renal and ovarian cancers, where CDH6 expression is prevalent [26].

This review provides an in-depth discussion of the advancements in DXd-based ADCs, their approved therapeutic applications, current clinical status, and future directions in this rapidly evolving field. We conducted a comprehensive literature search using the keyword “deruxtecan”, along with its brand and code names—T-DXd (DS-8201a, Enhertu^®^), Dato-DXd (DS-1062a, Datroway^®^), and P-DXd (U3-1402, HER3-DXd)—across the PubMed, Google Scholar, and ClinicalTrials.gov databases. The search included all preclinical and clinical studies published up to June 2025, focusing on full-text articles written in English. Clinical study data were retrieved from ClinicalTrials.gov using the same set of keywords, with particular attention paid to studies reporting outcome data.

## 2. Advantages, Structure, and Mechanism of Action of DXd-Based ADCs

### 2.1. Advantages of ADCs in Cancer Therapy

Compared to traditional chemotherapy, ADCs offer several advantages. First, ADCs enhance tumor specificity through antigen-targeting monoclonal antibodies, thereby reducing unintended toxicity [27]. Second, they enable the selective intracellular release of highly potent cytotoxic drugs, leading to improved therapeutic efficacy [28]. By ensuring targeted drug delivery, ADCs lower the required dose of systemic chemotherapy without compromising antitumor activity, thus contributing to a more favorable safety profile for patients [29].

Compared to other clinically established ADC platforms, DXd-based ADCs exhibit distinct structural and functional advantages. For example, MMAE-conjugated ADCs (e.g., brentuximab vedotin) and DM1-conjugated ADCs (e.g., T-DM1) are limited by non-cleavable linkers and poor membrane permeability, which restrict their bystander effect. In contrast, DXd-based ADCs incorporate cleavable linkers and membrane-permeable payloads, resulting in enhanced tumor penetration and cytotoxic diffusion. Nevertheless, head-to-head clinical trials directly comparing these platforms remain lacking, and conclusions regarding the superiority of one platform over another rely primarily on indirect or preclinical comparisons [2].

A comparative analysis of DXd-based ADCs and traditional chemotherapy is presented in Table 1.

### 2.2. Fundamental Structure of ADCs

ADCs are engineered biopharmaceuticals designed for the selective delivery of cytotoxic agents to tumor cells, thereby minimizing systemic toxicity. A conventional ADC comprises three fundamental components: a monoclonal antibody (mAb), a chemical linker, and a cytotoxic payload. Specifically, DXd-based ADCs consist of an mAb, a cleavable linker, and the topoisomerase I inhibitor payload DXd [30] (Figure 1A,B). The mAb recognizes and binds to specific tumor-associated antigens, facilitating targeted drug delivery. The linker is an essential element that attaches the payload to the antibody and governs its stability and controlled release [31]. The payload, typically a potent chemotherapeutic agent, exerts its cytotoxic effect upon internalization and release within the target cell [32].

### 2.3. Key Characteristics and Mechanism of DXd

DXd is a highly potent topoisomerase I inhibitor payload incorporated into next-generation ADCs [7]. Topoisomerase I inhibitors interfere with DNA replication by stabilizing the topoisomerase I–DNA cleavage complex, leading to DNA strand breaks and apoptosis in rapidly proliferating cancer cells [8]. DXd exhibits markedly enhanced membrane permeability compared to that of earlier ADC payloads, thereby amplifying the bystander effect. This unique feature enables DXd to exert cytotoxic activity not only in antigen-expressing tumor cells but also in adjacent tumor cells, effectively addressing tumor heterogeneity, a major challenge in cancer treatment [33]. Furthermore, DXd-based ADCs feature a high DAR and a stable, cleavable linker that enables efficient drug release upon internalization [34]. Upon binding to its target antigen, the ADC–receptor complex is internalized into the cell via receptor-mediated endocytosis [35]. Once internalized, the ADC undergoes progressive transport through early and late endosomes before reaching lysosomes. Within the lysosomal compartment, the acidic microenvironment and proteolytic enzymes facilitate linker degradation, resulting in the controlled release of a given cytotoxic payload such as DXd into the cytoplasm [36]. The released DXd then diffuses throughout the cytoplasm and enters the nucleus, where it inhibits topoisomerase I, thereby inducing DNA damage and promoting tumor cell death [34] (Figure 2). This mechanism ensures selective drug activation, thereby minimizing systemic toxicity while maximizing therapeutic efficacy [37].

## 3. Clinical Progress and Regulatory Status of DXd-Based ADCs

### 3.1. T-DXd (DS-8201a, Enhertu^®^)

T-DXd (Enhertu^®^) is a HER2-targeting ADC that is approved for multiple solid tumors, including breast, lung, and gastric cancers [38]. Initially granted accelerated approval on 20 December 2019 for the treatment of unresectable or metastatic HER2-positive breast cancer [12], its indications were later expanded to include HER2-low breast cancer [39]. This expansion represents a significant advancement in HER2-targeted therapy, allowing for treatment of tumors with low HER2 expression and broadening the clinical utility of HER2-directed ADCs [40]. Additionally, T-DXd has demonstrated efficacy in HER2-mutant NSCLC, leading to its U.S. FDA approval for this indication [41]. It is also approved for HER2-positive gastric and gastroesophageal junction (GEJ) adenocarcinoma based on clinical trial data indicating superior efficacy compared to that of conventional chemotherapy [42].

T-DXd consists of a humanized anti-HER2 monoclonal antibody conjugated to a membrane-permeable topoisomerase I inhibitor (DXd) via a cleavable GGFG tetrapeptide linker. This linker is selectively cleaved by lysosomal proteases, such as cathepsin B and L, within tumor cells. The DAR is approximately 8, which contributes to its potent bystander killing effect. This mechanism increases cytotoxicity against tumor cells with heterogeneous HER2 expression, making T-DXd particularly effective for treating tumors in complex microenvironments [34].

### 3.2. Dato-DXd (DS-1062a, Datroway^®^)

Dato-DXd (Datroway^®^) is a TROP2-directed ADC under investigation for multiple solid tumors, including TNBC, hormone receptor-positive (HR+) breast cancer, and NSCLC [43]. TROP2 is a transmembrane glycoprotein that is overexpressed in various malignancies and is associated with aggressive tumor progression and poor prognosis [44]. Early-phase clinical trials have demonstrated promising antitumor activity and a manageable safety profile for Dato-DXd in heavily pretreated patients [20]. Unlike conventional chemotherapy, Dato-DXd facilitates targeted delivery of a topoisomerase I inhibitor payload, potentially enhancing the therapeutic index while reducing systemic toxicity. Dato-DXd also employs a cleavable GGFG tetrapeptide linker, similar to that used in T-DXd, which enables selective cleavage by lysosomal proteases within tumor cells. It has a lower DAR of approximately 4, a design feature intended to balance therapeutic efficacy with reduced systemic toxicity due to TROP2 expression in some normal tissues [5].

On 17 January 2025, the FDA approved Dato-DXd for the treatment of adult patients with unresectable or metastatic HR-positive, HER2-negative breast cancer who had received prior endocrine-based therapy and chemotherapy [14]. Ongoing phase 3 trials will be crucial for determining its broader regulatory pathway and potential clinical applications for other cancer types [45].

### 3.3. P-DXd (U3-1402, HER3-DXd)

P-DXd (U3-1402) is an investigational HER3-targeting ADC designed for tumors overexpressing HER3, particularly epidermal growth factor receptor (EGFR)-mutant NSCLC [46]. HER3 is widely expressed in various solid tumors and plays a role in resistance to EGFR-targeted therapies [47]. Clinical trials have demonstrated promising efficacy of HER3-DXd in EGFR-mutant NSCLC patients who have developed resistance to tyrosine kinase inhibitors (TKIs), thus highlighting its potential as a novel therapeutic option [46]. Preclinical studies have suggested that P-DXd, like T-DXd, incorporates a cleavable GGFG tetrapeptide linker and may have a DAR of approximately 8, supporting its potent bystander effect [23].

In December 2023, the FDA granted priority review for the biologics license application (BLA) of P-DXd [48]. However, on 26 June 2024, the FDA issued a complete response letter (CRL) regarding the BLA submission of P-DXd, citing inspection deficiencies at a third-party manufacturing facility. Importantly, the CRL did not identify concerns regarding the drug’s efficacy or safety data [49]. Ongoing clinical trials aim to address these regulatory concerns while further evaluating the efficacy and safety profile of HER3-DXd in NSCLC and other malignancies [46].

## 4. Clinical Status of DXd-Based ADCs

There are currently numerous ongoing clinical trials for DXd-based ADCs. Only the results of trials with reported outcomes are summarized in Table 2.

The phase 2 trial NCT03248492 demonstrated the efficacy of T-DXd in HER2-positive metastatic breast cancer (mBC) patients who were previously treated with ado-trastuzumab emtansine (T-DM1, Kadcyla^®^). T-DXd exhibited substantial and sustained efficacy in these heavily pretreated patients, achieving a confirmed objective response rate (ORR) of 62%, a median duration of response (mDOR) of 18.2 months, and a median progression-free survival (mPFS) of 19.4 months [17,75,76,77,78,79,80,81].

The phase 2 trial NCT0448414 evaluated Dato-DXd in patients with advanced or metastatic NSCLC who had previously received treatments. The primary endpoint was ORR, with secondary endpoints including DOR, PFS, and overall survival (OS). Among 137 patients, 71.5% had received at least three prior treatments, and 56.9% possessed EGFR mutations. The confirmed ORR was 35.8%, with a median DOR of 7.0 months. Treatment-related adverse events of grade ≥ 3 occurred in 28.5% of the patients, with stomatitis being the most common. Interstitial lung disease (ILD)/pneumonitis occurred in 3.6%, along with one grade 5 event [131,132,133,134,135,136].

The phase 1/2 trial NCT02980341 evaluated P-DXd in patients with HER3-expressing mBC. The study reported ORR values of 30.1% in HR+/HER2-negative, 22.6% in TNBC, and 42.9% in HER2-positive patients, with mPFS values of 7.4, 5.5, and 11.0 months, respectively. The safety profile was manageable, although adverse events of grade ≥ 3, including ILD, occurred in 71.4% of the patients. These results support further investigation of P-DXd for breast cancer treatment [4,137,138,139,140,141].

## 5. Challenges and Clinical Applications of DXd-Based ADCs in Breast, Lung, and Gastric Cancers

Given the rapidly expanding clinical development of DXd-based ADCs, a focused analysis on cancer types with the most robust data is essential for understanding their therapeutic impact and translational potential [60]. The clinical success of DXd-based ADCs has been most evident in three major solid tumor types: breast cancer, lung cancer, and gastric cancer [17,60,144]. These malignancies represent the most advanced indications for DXd-based ADCs in terms of regulatory approval, clinical trial activity, and therapeutic efficacy [145]. This section summarizes the therapeutic implications and clinical progress of DXd-based ADCs in these key cancer types. Their molecular targets, clinical development stages, and key findings in these tumor types are summarized in Table 3.

### 5.1. Breast Cancer

Breast cancer has been a primary focus in the development of DXd-based ADCs, particularly T-DXd (Enhertu^®^). Initially approved for HER2-positive metastatic breast cancer, T-DXd’s indication has since been expanded to include HER2-low-expressing tumors, signifying a paradigm shift in HER2-targeted therapy [40]. In the DESTINY-Breast04 trial, T-DXd significantly improved progression-free survival and overall survival in patients with HER2-low metastatic breast cancer, leading to its U.S. FDA approval for this indication [107].

Dato-DXd (Datroway^®^), a TROP2-targeted ADC, is under investigation in heavily pretreated HR+/HER2-negative and TNBC patients. The TROPION-PanTumor01 study showed encouraging antitumor activity and a manageable safety profile, further supporting the clinical utility of DXd-based ADCs in breast cancer [20].

In addition to T-DXd, P-DXd has shown promise in clinical trials for HER3-expressing breast cancers, including HR+/HER2-negative, TNBC, and HER2-positive subtypes. The phase 1/2 study NCT02980341 reported ORR of 30.1% in HR+/HER2-negative, 22.6% in TNBC, and 42.9% in HER2-positive patients [4].

### 5.2. Lung Cancer

In NSCLC, several DXd-based ADCs have entered clinical development [146]. T-DXd (Enhertu^®^) has shown strong efficacy in HER2-mutant NSCLC [60]. The DESTINY-Lung01 and DESTINY-Lung02 trials demonstrated a high objective response rate and durable clinical benefit, leading to its FDA approval for HER2-mutated metastatic NSCLC [51,59].

Dato-DXd, which targets TROP2, has also shown promise in NSCLC. In the phase 2 trial NCT04484142, a confirmed ORR of 35.8% was achieved in patients with advanced or metastatic NSCLC, many of whom had received three or more prior treatments [136].

P-DXd has been evaluated in EGFR-mutant NSCLC patients who developed resistance to TKIs. The HERTHENA-Lung01 trial reported favorable efficacy and a manageable safety profile, highlighting P-DXd’s potential in this difficult-to-treat population [23].

### 5.3. Gastric Cancer

T-DXd (Enhertu^®^) has been approved for HER2-positive advanced gastric or GEJ adenocarcinoma. The DESTINY-Gastric01 trial demonstrated a significant improvement in overall survival compared to standard chemotherapy in Japanese and Korean patients who had progressed on prior trastuzumab-based regimens [144].

Subsequent trials, such as DESTINY-Gastric02 and DESTINY-Gastric06, have expanded investigations into Western populations and earlier lines of treatment. These findings underscore T-DXd (Enhertu^®^)’s potential to address the unmet needs in gastric cancer treatment, particularly in patients with HER2-positive disease [53,69].

### 5.4. Challenges in Expanding DXd-Based ADCs to Other Cancers

While DXd-based ADCs have demonstrated remarkable efficacy in breast, lung, and gastric cancers [12,14,41,42], their success in other tumor types remains limited. Several investigational agents, such as I-DXd, targeting B7-H3 [24], and R-DXd, targeting CDH6, are being explored in prostate, ovarian, and renal cancers. However, these efforts have yet to yield regulatory approvals. This discrepancy may be attributed to several factors: (i) lower or heterogeneous expression of ADC targets in non-traditional tumor types, (ii) increased risk of systemic toxicity due to off-tumor target expression, and (iii) a lack of robust preclinical models that accurately reflect human disease biology. Therefore, further target validation, biomarker-guided patient selection, and optimized ADC design are essential to expand the clinical utility of DXd-based ADCs beyond currently approved indications [26].

## 6. Challenges and Future Directions of DXd-Based ADCs

### 6.1. Resistance Mechanisms to DXd-Based ADCs

While DXd-based ADCs have demonstrated notable success in several malignancies, including breast, lung, and gastric cancers, their clinical deployment still faces important limitations that warrant further investigation. Despite the promising efficacy of DXd-based ADCs, resistance remains a significant challenge that can be categorized into intrinsic and acquired forms [147].

Intrinsic resistance arises from low or heterogeneous expression of target antigens (HER2, HER3, and TROP2) in tumors, which limits ADC effectiveness by reducing sufficient target binding [148]. Additionally, alterations in endocytic pathways or lysosomal dysfunctions can impede ADC internalization and payload release, thus decreasing therapeutic efficacy [149].

Acquired resistance, which develops after treatment exposure, is often associated with the upregulation of efflux pumps such as ATP-binding cassette (ABC) transporters that export the cytotoxic payload, thereby reducing drug effectiveness [150]. Moreover, the upregulation of DNA repair pathways enables cancer cells to repair damage caused by topoisomerase I inhibition [6]. Mutations in target antigens, such as HER2 mutations or HER3 overexpression, may drive resistance in tumors that initially responded but later progressed [151].

Ongoing research is focused on addressing resistance through novel ADC formulations and combination therapies, including dual-targeted ADCs or bispecific antibodies, which aim to overcome antigen heterogeneity [152]. Combination strategies with targeted agents, immune checkpoint inhibitors (ICIs), or chemotherapy are also being explored to counteract resistance and improve therapeutic outcomes [153]. For example, recent clinical studies have explored the combination of T-DXd with immune checkpoint inhibitors to enhance antitumor immunity [154]. In addition, combining T-DXd with PARP inhibitors such as olaparib is under investigation to overcome resistance associated with DNA repair pathways [155]. Circulating tumor DNA (ctDNA)-based monitoring strategies are also being examined as tools to predict and track emerging resistance mechanisms [154,155].

### 6.2. Toxicity and Safety Concerns

DXd-based ADCs—including T-DXd, Dato-DXd, and P-DXd—have demonstrated significant therapeutic efficacy; however, their use is associated with various adverse events that warrant careful consideration. ILD is a particularly adverse event observed with T-DXd, leading to severe respiratory issues and potentially fatal outcomes [156]. In clinical trials, ILD occurred in approximately 13.6% of patients treated with T-DXd, with some fatal cases reported [157]. Similarly, ILD has also been reported in patients receiving Dato-DXd, necessitating vigilant monitoring [158]. Mechanistically, pulmonary toxicities such as ILD may result from immune-mediated inflammatory responses, as well as non-specific uptake of ADCs in lung tissue, off-target effects due to low-level target expression, or the systemic diffusion of cytotoxic payloads following premature cleavage of linkers. Although the precise etiology remains under investigation, these factors are considered key contributors to ILD pathogenesis in patients receiving T-DXd and Dato-DXd [156,157].

Common hematologic toxicities associated with these ADCs include neutropenia and anemia [159]. For instance, in the DESTINY-Breast01 trial, grade 3 or higher neutropenia occurred in 20.7% of patients receiving T-DXd [160]. Patients treated with these ADCs also frequently experience gastrointestinal side effects such as nausea and vomiting [161]. In the DESTINY-Breast01 trial, nausea of any grade was reported in over 70% of patients receiving T-DXd, leading to its classification as a highly emetogenic agent [76]. Hematologic toxicities, particularly neutropenia and anemia, are common dose-limiting adverse events associated with DXd-based ADCs. These effects are thought to arise primarily from the off-target uptake of the cytotoxic payload by proliferative hematopoietic precursor cells in the bone marrow. Membrane-permeable payloads such as DXd can diffuse into these non-target cells, impairing myelopoiesis and erythropoiesis, which leads to reduced neutrophil and red blood cell counts [162].

Elevated liver enzymes have been observed in some patients treated with T-DXd, indicating potential hepatotoxicity [163]. Hepatotoxicity associated with ADCs may result from off-target uptake by hepatocytes or Kupffer cells through Fc receptor-mediated endocytosis or non-specific pinocytosis. In particular, the liver’s active role in metabolism and clearance makes it susceptible to toxic payload accumulation, potentially leading to hepatocellular injury and enzyme elevation [162].

Additionally, decreased left ventricular ejection fraction (LVEF) has been reported in patients treated with T-DXd. In the DESTINY-Breast03 trial, 2.3% of the patients experienced LVEF reduction, all of which were asymptomatic and mostly resolved without intervention [164]. Cardiotoxicity may be attributed to the expression of low levels of HER2 in cardiac tissue, leading to on-target, off-tumor toxicity [157]. Moreover, the systemic release of cytotoxic payloads may also contribute to myocardial injury through direct mitochondrial or oxidative stress pathways [162,164]. This is of particular concern in patients with pre-existing cardiac conditions or prior anthracycline exposure [164].

To mitigate these adverse effects, ongoing clinical trials are investigating lower doses, alternative dosing schedules, and the development of next-generation ADCs with improved safety profiles [165]. Additionally, the use of predictive biomarkers to identify patients at risk for severe toxicities may help to tailor treatment regimens and minimize adverse events [149].

### 6.3. Strategies to Overcome Challenges in DXd-Based ADCs

To address limitations such as resistance from antigen heterogeneity and off-target toxicity in DXd-based ADCs, several structure-based and clinical strategies are under investigation [154,162,165,166,167]. One approach involves modifying linker chemistry, payload properties, and drug-to-antibody ratios to optimize the therapeutic index. Cleavable linkers—such as the maleimidocaproyl-Gly-Gly-Phe-Gly (MC-GGFG) linker in T-DXd—allow enzymatic cleavage within the lysosome and subsequent extracellular diffusion of membrane-permeable payloads, contributing to the bystander effect. In contrast, non-cleavable linkers—such as succinimidyl 4-(*N*-maleimidomethyl)cyclohexane-1-carboxylate (SMCC) in T-DM1—result in intracellular retention of payload catabolites, thereby limiting bystander killing. While non-cleavable linkers may help reduce systemic toxicity, they can compromise efficacy in heterogeneous tumors due to insufficient payload diffusion into antigen-negative tumor cells. These design trade-offs—between safety and bystander activity—are being actively optimized in the development of next-generation ADCs [166,167].

Despite these structural refinements, ADCs still face additional hurdles in achieving effective tumor penetration, particularly in solid malignancies. Another key obstacle for ADC delivery in solid tumors is the binding site barrier (BSB), which arises when ADCs preferentially bind to antigen-expressing tumor cells located near the vasculature, thereby limiting deeper tissue penetration and leading to suboptimal intratumoral distribution. This phenomenon is particularly pronounced in tumors with high antigen density and poor vascularization [168]. Strategies to overcome BSB include modifying antibody affinity, reducing ADC molecular size, employing biparatopic or bispecific constructs, and using site-specific conjugation techniques to improve tumor penetration [169]. However, even if physical delivery barriers such as the BSB are addressed, DXd-based ADCs must still overcome critical biological resistance mechanisms—including antigen heterogeneity, drug efflux, and enhanced DNA repair—which significantly limit long-term efficacy [170]. To address both physical and biological resistance mechanisms in DXd-based ADCs, various preclinical and clinical efforts are underway to develop combination strategies [171]. These include the use of immune checkpoint inhibitors (ICIs), PARP inhibitors, and rational modifications to ADC design [154].

Combination therapies with ICIs or resistance pathway inhibitors may enhance tumor response and overcome resistance [154]. For example, combining DXd-based ADCs with programmed cell death protein 1 (PD-1)/programmed cell death ligand 1 (PD-L1) inhibitors or DNA repair pathway inhibitors has shown promise in overcoming resistance mechanisms [147,155]. Predictive biomarkers such as circulating tumor DNA and transcriptomic analyses are being explored to enable early detection of tumor response, monitor resistance mechanisms, and personalize therapeutic regimens [149]. Studies have demonstrated that ctDNA can reflect treatment efficacy and resistance mutations in real time, allowing for adaptive treatment strategies [172]. Additionally, dose modifications and alternative administration schedules are being evaluated clinically to reduce toxicity while maintaining efficacy [165]. Ongoing clinical trials are investigating lower doses and more flexible dosing schedules to improve safety without compromising therapeutic benefit [162].

Alongside these therapeutic advancements, scaling the manufacturing process of DXd-based ADCs presents another key challenge for clinical translation [173]. The complexity of ADC production—particularly achieving consistent conjugation efficiency, DAR, and linker stability—poses significant technical and regulatory hurdles [2]. Process standardization and advances in site-specific conjugation technologies are being actively pursued to ensure batch-to-batch consistency, reduce production costs, and improve commercial scalability [174]. Future innovations in ADC manufacturing, including automation and continuous bioprocessing, may further streamline production and facilitate global accessibility [175].

In summary, further enhancement of DXd-based ADCs may involve developing optimized cleavable linkers with high enzymatic specificity [176], exploring alternative cytotoxic payloads (e.g., DNA alkylating agents or tubulin inhibitors) [2], and refining patient stratification through predictive biomarkers and genomic profiling [177]. These evolving strategies aim to personalize ADC therapies and maximize clinical benefit while minimizing adverse effects [2]. In light of these multifaceted efforts, continued research is essential to further optimize the design and application of DXd-based ADCs in oncology—particularly to overcome resistance mechanisms such as antigen heterogeneity, efflux pump activity, and DNA repair upregulation [148,178]. Mechanistically grounded strategies, including advanced ADC engineering, rationally designed combination therapies, and biomarker-guided patient selection, will be critical for improving therapeutic outcomes and expanding the clinical utility of these promising agents [155,179,180].

## 7. Conclusions

DXd-based ADCs represent a transformative advancement in targeted cancer therapy, combining potent cytotoxic payloads with the specificity of antibodies. Their success in treating HER2-, HER3-, and TROP2-expressing tumors—as demonstrated by FDA approvals and promising clinical trial outcomes—underscores their potential to improve patient outcomes. The high drug-to-antibody ratio, stable linker system, and bystander effect contribute to their superior efficacy, addressing challenges such as tumor heterogeneity and resistance. Despite these advantages, challenges related to toxicity, resistance mechanisms, and manufacturing complexities persist. Future research should focus on optimizing patient selection using biomarker-driven strategies, refining ADC design to enhance both safety and efficacy, and exploring novel combinations with immunotherapy and targeted agents. Ongoing clinical investigations will be critical for defining the role of DXd-based ADCs across various malignancies, ultimately expanding their therapeutic applications and reinforcing their position as a cornerstone of modern oncology.

## Figures and Tables

**Figure 1 ijms-26-06523-f001:**
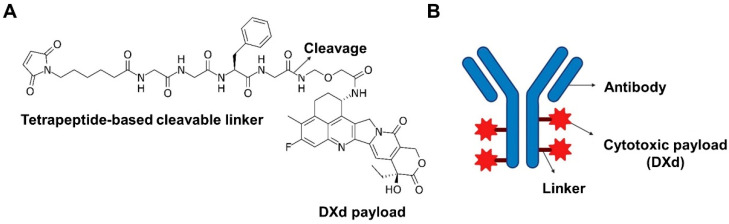
Chemical structure of DXd and schematic design of a DXd-based ADC. (**A**) Chemical structure of DXd, a topoisomerase I inhibitor derived from exatecan, with the conjugation site to a tetrapeptide-based cleavable linker clearly indicated. The diagram includes annotations such as “cleavage” and “DXd payload” to denote the enzymatic cleavage site of the linker and the subsequent release point of the active cytotoxic component within the lysosomal compartment of tumor cells. This chemical structure represents an optimized design that enables selective drug release specifically in the tumor microenvironment or lysosome. (**B**) This schematic illustration depicts a DXd-based ADC composed of a monoclonal antibody (mAb) that selectively targets tumor-associated antigens, a cleavable linker that remains stable in systemic circulation but undergoes enzymatic degradation within the tumor microenvironment or lysosome, and the cytotoxic payload DXd. Upon intracellular release, DXd induces cancer cell death. Figure created with BioRender.com. ADC, antibody–drug conjugate; DXd, deruxtecan; mAb, monoclonal antibody.

**Figure 2 ijms-26-06523-f002:**
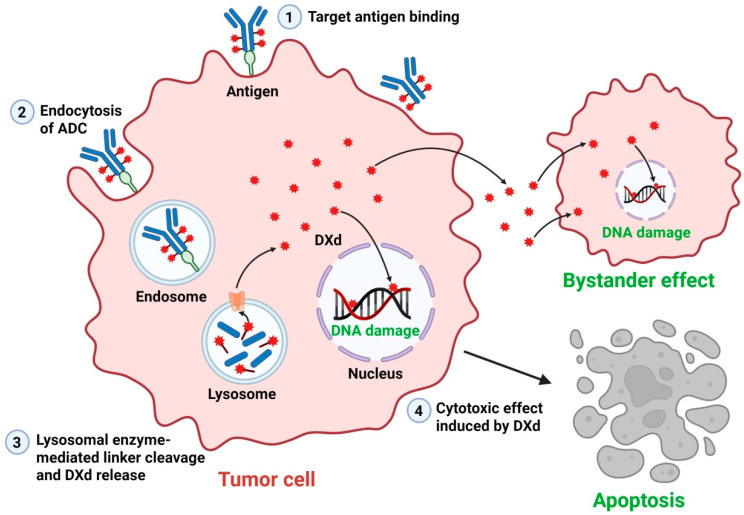
Mechanism of action of DXd-based ADCs. DXd-based ADCs selectively bind to target antigens expressed on the surface of tumor cells (1). The ADC–antigen complex is internalized via receptor-mediated endocytosis (2) and subsequently trafficked to the lysosome (3), where the cleavable linker is enzymatically degraded by lysosomal proteases, leading to the intracellular release of the topoisomerase I inhibitor DXd. The released DXd then translocates into the nucleus, where it inhibits topoisomerase I, induces DNA damage, and subsequently triggers apoptosis (4). In addition, due to its high membrane permeability, DXd can diffuse into neighboring antigen-negative tumor cells, eliciting a bystander cytotoxic effect that enhances the overall antitumor response. Figure created with BioRender.com. ADCs, antibody–drug conjugates; DXd, deruxtecan.

**Table 1 ijms-26-06523-t001:** Comparison of DXd-based ADCs and traditional chemotherapy.

Feature	DXd-Based ADCs	Traditional Chemotherapy
Targeting mechanism	High specificity via antigen-targeting monoclonal antibodies	Non-specific distribution throughout the body
Mechanism of action	Topoisomerase I inhibition via membrane-permeable DXd payload	Various (e.g., DNA intercalation, alkylation, microtubule inhibition)
Delivery method	ADC administration via IV	IV or oral administration
Selectivity	High (tumor antigen-dependent)	Low (affects both cancerous and healthy proliferating cells)
Systemic toxicity	Lower in some patients; ILD and neutropenia remain significant concerns	High; damages rapidly dividing normal tissues (e.g., GI, bone marrow)
Bystander effect	Present due to cleavable linker and membrane-permeable payload	Not applicable; traditional chemotherapies act systemically and non-selectively without relying on local diffusion mechanisms
Half-life	Prolonged due to antibody component	Shorter; depends on formulation and metabolism
Common side effects	Neutropenia, nausea, fatigue, vomiting, ILD; rare cardiotoxicity	Neutropenia, mucositis, alopecia, nausea, vomiting, cardiotoxicity
Therapeutic index	Higher in selected patients; influenced by antigen expression	Lower; narrow margin between efficacy and toxicity
Clinical applications	Approved or under investigation in HER2-, HER3-, and TROP2-positive breast, lung, gastric, and colorectal cancers	Broad-spectrum use in various cancers
Regulatory status	FDA-approved: T-DXd (DS-8201a; Enhertu^®^), Dato-DXd (DS-1062a; Datroway^®^); investigational: P-DXd (U3-1402, HER3-DXd)	Widely approved as standard treatment in oncology

ADCs, antibody–drug conjugates; Dato-DXd, datopotamab deruxtecan, DS-1062a, Datroway^®^; DXd, deruxtecan; FDA, Food and Drug Administration; GI, gastrointestinal; HER2, human epidermal growth factor receptor 2; HER3, human epidermal growth factor receptor 3; ILD, interstitial lung disease; IV, intravenous; P-DXd, patritumab deruxtecan, U3-1402, HER3-DXd; T-DXd, trastuzumab deruxtecan, DS-8201a, Enhertu^®^; TROP2, trophoblast cell surface antigen 2.

**Table 2 ijms-26-06523-t002:** Overview of clinical trials for DXd-based ADCs.

Interventions	Study Title	Conditions	Phase	NCT Number	Refs.
T-DXd	T-DXd in participants with HER2-mutated metastatic NSCLC (DESTINY-LUNG02)	NSCLC	Phase 2	NCT04644237	[50,51]
T-DXd	Study of T-DXd monotherapy in patients with HER2-expressing locally advanced or metastatic gastric or GEJ adenocarcinoma who have received 2 or more prior regimens (DG-06)	GEJ adenocarcinoma	Phase 2	NCT04989816	[52,53]
T-DXd	A study of T-DXd for the treatment of solid tumors harboring HER2 activating mutations (DPT01)	Advanced solid tumors with HER2 mutation	Phase 2	NCT04639219	[54,55,56]
T-DXd	A single arm phase 2 study to evaluate efficacy and safety of trastuzumab DXd for patients with HER2 mutant NSCLC (DL-05)	HER2-mutant NSCLC	Phase 2	NCT05246514	[57,58]
T-DXd	DS-8201a in HER2-expressing or -mutated NSCLC (DESTINY-Lung01)	NSCLC	Phase 2	NCT03505710	[59,60,61,62,63,64]
T-DXd	DS-8201a in HER2-positive gastric cancer that cannot be surgically removed or has spread (DESTINY-Gastric02)	Adenocarcinoma gastric stage IV with metastases, Adenocarcinoma—GEJ	Phase 2	NCT04014075	[65,66,67,68,69]
DS-8201a	DS-8201a in HER2-expressing colorectal cancer (DESTINY-CRC01)	Colorectal neoplasm	Phase 2	NCT03384940	[70,71,72,73,74]
DS-8201a	A study of DS-8201a in metastatic breast cancer previously treated with T-DM1	Breast cancer	Phase 2	NCT03248492	[17,75,76,77,78,79,80,81]
DS-8201a	DS-8201a in patients with cancer that tests positive for HER2 protein	Gastric adenocarcinoma, Breast neoplasm	Phase 1	NCT03368196	[82,83]
DS-8201a	Phase 1 study to evaluate the effect of DS-8201a on the QT/QTc interval and pharmacokinetics in HER2-expressing breast cancer	Malignant neoplasm of breast	Phase 1	NCT03366428	[84]
DS-8201a	T-DXd in participants with HER2-overexpressing advanced or metastatic colorectal cancer (DESTINY-CRC02)	Advanced colorectal cancer	Phase 2	NCT04744831	[85,86]
DS-8201a	Study of DS-8201a in subjects with advanced solid malignant tumors	Advanced solid tumors	Phase 1	NCT02564900	[79,87,88,89,90,91,92]
T-DXd, Nivolumab	T-DXd with nivolumab in advanced breast and urothelial cancer	Breast cancer, Urothelial carcinoma	Phase 1	NCT03523572	[93,94,95,96,97,98]
T-DXd, SBT6050, Tucatinib, Trastuzumab, Capecitabine	A safety and activity study of SBT6050 in combination with other HER2-directed therapies for HER2-positive cancers.	HER2-positive breast cancer, HER2-positive gastric cancer, HER2-positive colorectal cancer, HER2-expressing NSCLC	Phase 1, Phase 2	NCT05091528	[99]
T-DXd, Cape citabine, Lapatinib, Trastuzumab	DS-8201a in pre-treated HER2 breast cancer that cannot be surgically removed or has spread [DESTINY-Breast02]	Breast cancer	Phase 3	NCT03523585	[100,101,102,103,104,105]
T-DXd (DS-8201a), Capecitabine, Eribulin, Gemcitabine, Paclitaxel, Nab-paclitaxel	T-DXd (DS-8201a) versus investigator’s choice for HER2-low breast cancer that has spread or cannot be surgically removed [DESTINY-Breast04]	Breast cancer	Phase 3	NCT03734029	[106,107,108,109,110,111,112,113]
T-DXd, T-DM1	DS-8201a versus T-DM1 for HER2-positive, unresectable and/or metastatic breast cancer previously treated with trastuzumab and taxane [DESTINY-Breast03]	Breast cancer	Phase 3	NCT03529110	[76,114,115,116,117,118,119]
DS-8201a, Physician’s choice	DS-8201a in HER2-expressing gastric cancer [DESTINY-Gastric01]	Gastrointestinal neoplasm	Phase 2	NCT03329690	[120,121,122,123,124]
DS-8201a, Ritonavir, Itraconazole	Study of DS-8201a for participants with advanced solid malignant tumors	Neoplasm metastasis	Phase 1	NCT03383692	[125,126,127,128,129,130]
DS-1062a	Study of DS-1062a in advanced or metastatic NSCLC with actionable genomic alterations (TROPION-Lung05)	NSCLC	Phase 2	NCT04484142	[131,132,133,134,135,136]
P-DXd	Phase I/II study of U3-1402 in subjects with HER3 positive metastatic breast cancer	Metastatic breast cancer	Phase 1, Phase 2	NCT02980341	[4,137,138,139,140,141]
P-DXd	HERTHENA-Lung01: P-DXd in subjects with metastatic or locally advanced EGFR-mutated NSCLC	NSCLC metastatic, NSCLC mutation in HER	Phase 2	NCT04619004	[23,142,143]

DS-1062a, datopotamab deruxtecan, Dato-DXd, Datroway^®^; GEJ, gastroesophageal junction; HER, human epidermal growth factor receptor; NSCLC, non-small cell lung cancer; P-DXd, patritumab deruxtecan, U3-1402, HER3-DXd; T-DM1, ado-trastuzumab emtansine, Kadcyla^®^; T-DXd, trastuzumab deruxtecan, DS-8201a, Enhertu^®^.

**Table 3 ijms-26-06523-t003:** Comparative clinical profiles of key DXd-based ADCs across major cancer (breast cancer, lung cancer, stomach cancer) types.

Cancer Type	ADC Agent	Target	Clinical Phase/Approval	Key Findings	Refs.
Breast Cancer	T-DXd	HER2	FDA Approved (HER2+/HER2-low)	Improved PFS and OS in HER2-low mBC	[107]
	Dato-DXd	TROP2	FDA Approved (HR+/HER2–mBC)	Promising efficacy in TNBC and HR+/HER2–mBC	[20]
	P-DXd	HER3	Phase 1/2	ORR: 22.6% (TNBC), 30.1% (HR+/HER2–mBC)	[4]
Lung Cancer	T-DXd	HER2	FDA Approved (HER2-mutant NSCLC)	High ORR and durable response	[51,59]
	Dato-DXd	TROP2	Phase 2	ORR 35.8% in heavily pretreated NSCLC	[136]
	P-DXd	HER3	Phase 2	Active in EGFR-mutant, TKI-resistant NSCLC	[23]
Gastric Cancer	T-DXd	HER2	FDA Approved (HER2+ gastric cancer)	Improved OS vs. chemotherapy in DESTINY-Gastric01	[144]

ADC, antibody–drug conjugate; Dato-DXd, datopotamab deruxtecan, DS-1062a, Datroway^®^; EGFR, epidermal growth factor receptor; FDA, Food and Drug Administration; HER2, human epidermal growth factor receptor 2; HER3, human epidermal growth factor receptor 3; HR+, hormone receptor-positive; mBC, metastatic breast cancer; NSCLC, non-small cell lung cancer; ORR, objective response rate; OS, overall survival; P-DXd, patritumab deruxtecan, U3-1402, HER3-DXd; PFS, progression-free survival; TKI, tyrosine kinase inhibitor; T-DXd, trastuzumab deruxtecan, DS-8201a, Enhertu^®^; TROP2, trophoblast cell surface antigen 2; TNBC, triple-negative breast cancer.

## Data Availability

Data presented in this study are available upon request from the corresponding author.

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
