# Peer review of "Antibody–Drug Conjugates Powered by Deruxtecan: Innovations and Challenges in Oncology"

_ijms, 2025, doi:10.3390/ijms26136523_

Round 1
Reviewer 1 Report (New Reviewer)
Comments and Suggestions for Authors
This article presents clinical data on an antibody-drug conjugate (ADC) with deruxtecan (DXd) as the payload and is a highly informative review article for clinicians and researchers involved in ADC development.
However, this article raises several questions.
1) Some ADCs with DXd as the payload have completed clinical evaluations and been approved by the FDA, but not all of them are performing well. For example, DS-6157a, a conjugate of an antibody against G protein-coupled receptor (GPCR) and DXd, has halted development in phase 1, but the description gives rise to expectations.
It is advisable to avoid descriptions that give the general reader the impression that all DXd-ADCs are successful. This is not necessarily beneficial for DXd-ADCs that are currently successful.
2) The toxicity of this ADC, particularly interstitial pneumonia, is well known among clinicians and is described in section 6.2, “Toxicity and Safety Concerns,” of this article. However, in Table 1, under “Systemic Toxicity,” it is stated that DXd-ADC has “generally lower” toxicity, but only mentions that ILD has been reported. I believe it would be more accurate to omit “generally lower.”
Author Response
Response to Reviewer 1 Comments
Dear editor Natthasit Chanthachitpreecha and reviewers:
Thank you for your letter and for the reviewers’ comments concerning our manuscript entitled “Antibody-Drug Conjugates Powered by Deruxtecan: Innovations and Challenges in Oncology” (ID: ijms-3723876). These comments are all valuable and very helpful for revising and improving our paper, as well as the important guiding significance to our researches. We have studied comments carefully and have made corrections which we hope meet with approval. Revised portions are marked in red on the paper. The main corrections in the paper and the response to the reviewer’s comments are as flowing:
- Point 1: Some ADCs with DXd as the payload have completed clinical evaluations and been approved by the FDA, but not all of them are performing well. For example, DS-6157a, a conjugate of an antibody against G protein-coupled receptor (GPCR) and DXd, has halted development in phase 1, but the description gives rise to expectations.
It is advisable to avoid descriptions that give the general reader the impression that all DXd-ADCs are successful. This is not necessarily beneficial for DXd-ADCs that are currently successful.
- Response 1: We appreciate the reviewer’s insightful comment. We have revised the text in Section 1 (Introduction) to clarify that while several DXd-based ADCs have shown promising clinical outcomes, not all have advanced successfully. In particular, we now mention that DS-6157a development was discontinued in Phase 1 due to limited efficacy, reflecting the variability in performance among DXd-ADCs.
- Point 2: The toxicity of this ADC, particularly interstitial pneumonia, is well known among clinicians and is described in section 6.2, “Toxicity and Safety Concerns,” of this article. However, in Table 1, under “Systemic Toxicity,” it is stated that DXd-ADC has “generally lower” toxicity, but only mentions that ILD has been reported. I believe it would be more accurate to omit “generally lower.”
- Response 2: Thank you for pointing this out. We have modified the wording in Table 1 under “Systemic Toxicity” to: “Lower in some patients; ILD and neutropenia remain significant concerns.” This phrasing provides a more balanced view of the toxicity profile.
We tried our best to improve the manuscript and made some changes in the manuscript. These changes will not influence the content and framework of the paper. And here we did not list the changes but marked them in red in the revised paper.
We appreciate for Editors/Reviewers’ warm work earnestly and hope that the correction will meet with approval.
Once again, thank you very much for your comments and suggestions.

Reviewer 2 Report (New Reviewer)
Comments and Suggestions for Authors
The manuscript is a comprehensive and well-referenced review that discusses the development, mechanism of action, clinical progress, and challenges of deruxtecan (DXd)-based antibody–drug conjugates (ADCs) in oncology. It covers approved therapies (e.g., T-DXd, Dato-DXd), ongoing clinical trials, resistance mechanisms, safety concerns (notably interstitial lung disease), and future directions including biomarker-guided personalization.
The review can be accepted for publication in the journal after minor modifications are made.
- The authors focused on the benefits of described ADC. Please include a more balanced comparison between DXd-based ADCs and other ADC platforms. Briefly comment on the lack of head-to-head clinical trials with alternative payloads or linker strategies.
- Please add a graphical abstract summarizing DXd-based ADC structure and clinical targets.
- Please define the abbreviations at first use (e.g., DAR, ILD).
Author Response
Response to Reviewer 2 Comments
Dear editor Natthasit Chanthachitpreecha and reviewers:
Thank you for your letter and for the reviewers’ comments concerning our manuscript entitled “Antibody-Drug Conjugates Powered by Deruxtecan: Innovations and Challenges in Oncology” (ID: ijms-3723876). These comments are all valuable and very helpful for revising and improving our paper, as well as the important guiding significance to our researches. We have studied comments carefully and have made corrections which we hope meet with approval. Revised portions are marked in red on the paper. The main corrections in the paper and the response to the reviewer’s comments are as flowing:
- Point 1: The authors focused on the benefits of described ADC. Please include a more balanced comparison between DXd-based ADCs and other ADC platforms. Briefly comment on the lack of head-to-head clinical trials with alternative payloads or linker strategies.
- Response 1: We appreciate the reviewer’s insightful comment. We have revised Section 2.1 to incorporate a more balanced comparison between DXd-based ADCs and other ADC platforms, such as MMAE- and DM1-conjugated ADCs. We have also added a note highlighting the current lack of head-to-head clinical trials comparing different payloads or linker strategies. These additions provide a more comprehensive context for interpreting the clinical value of DXd-based ADCs.
- Point 2: Please add a graphical abstract summarizing DXd-based ADC structure and clinical targets.
- Response 2: Thank you for your comments. We have revised the previous graphical abstract summarizing the key structural features and clinical targets of DXd-based ADCs.
- Point 3: Please define the abbreviations at first use (e.g., DAR, ILD).
- Response 3: Thank you for your comments. In the manuscript, DAR is defined on page 2, line 54, and ILD is defined on page 8, line 262. We have carefully reviewed the manuscript to ensure that all abbreviations are defined when they first appear in the text. We have also defined abbreviations separately on page 14.
We tried our best to improve the manuscript and made some changes in the manuscript. These changes will not influence the content and framework of the paper. And here we did not list the changes but marked them in red in the revised paper.
We appreciate for Editors/Reviewers’ warm work earnestly and hope that the correction will meet with approval.
Once again, thank you very much for your comments and suggestions.

Reviewer 3 Report (New Reviewer)
Comments and Suggestions for Authors
In the manuscript, the authors provide a comprehensive review of DXd-based ADCs, their clinical progress, challenges and strategies to overcome the limitations. I have a few comments and suggestions for the authors.
For section 3, please provide information for each ADC’s structure, such as the type of linker, mechanism for linker cleavage, and the DAR value.
For section 5, please add some comments on the status of ADC application in other types of cancers. If fewer investigations of using ADC for the treatment of cancers other than breast, lung, and gastric cancers, or a lower success rate for those studies, what would be the reasons?
Lines 332-336: Please provide some examples of the ongoing research focusing on addressing resistance that are related to DXd-based ADCs.
For section 6.2, please provide more information about the mechanisms of ADC toxicity, as they are relevant to the strategies to overcome.
The binding site barrier (BSB) has been proposed to be a challenge for ADCs in treating solid tumors. Please discuss this challenge.
Author Response
Response to Reviewer 3 Comments
Dear editor Natthasit Chanthachitpreecha and reviewers:
Thank you for your letter and for the reviewers’ comments concerning our manuscript entitled “Antibody-Drug Conjugates Powered by Deruxtecan: Innovations and Challenges in Oncology” (ID: ijms-3723876). These comments are all valuable and very helpful for revising and improving our paper, as well as the important guiding significance to our researches. We have studied comments carefully and have made corrections which we hope meet with approval. Revised portions are marked in red on the paper. The main corrections in the paper and the response to the reviewer’s comments are as flowing:
- Point 1: For section 3, please provide information for each ADC’s structure, such as the type of linker, mechanism for linker cleavage, and the DAR value.
- Response 1: We thank the reviewer for their insightful comment. In response, we have revised Section 3 to include detailed structural information for each DXd-based ADC. Specifically, we have now clearly stated the type of linker, its mechanism of cleavage (lysosomal protease-mediated), and the drug-to-antibody ratio (DAR) for each agent (T-DXd, Dato-DXd, and P-DXd).
- Point 2: For section 5, please add some comments on the status of ADC application in other types of cancers. If fewer investigations of using ADC for the treatment of cancers other than breast, lung, and gastric cancers, or a lower success rate for those studies, what would be the reasons?
- Response 2: We appreciate the reviewer’s thoughtful comment regarding the limited application of DXd-based ADCs beyond breast, lung, and gastric cancers. In response, we have added a new subsection (Section 5.4) titled “Challenges in Expanding DXd-Based ADCs to Other Cancers”, which discusses the current status of investigational agents such as ifinatamab deruxtecan (I-DXd) and raludotatug deruxtecan (R-DXd). This section addresses the relative lack of success and regulatory approvals in other tumor types, and explores potential contributing factors such as target heterogeneity, off-tumor toxicity, and limitations of preclinical models. These additions provide a more balanced and comprehensive overview of the translational barriers to ADC development in diverse cancer settings.
- Point 3: Lines 332-336: Please provide some examples of the ongoing research focusing on addressing resistance that are related to DXd-based ADCs.
- Response 3: We thank the reviewer for this valuable suggestion. In response, we have revised Section 6.1 to include representative examples of ongoing research addressing resistance to DXd-based ADCs. Specifically, we have added descriptions of clinical strategies involving the combination of DXd-based ADCs with immune checkpoint inhibitors and PARP inhibitors, which are currently under investigation to enhance therapeutic efficacy and overcome resistance mechanisms. We also included the emerging role of circulating tumor DNA (ctDNA) as a noninvasive biomarker to monitor resistance development. These additions strengthen the section by providing concrete and up-to-date examples of translational efforts focused on overcoming resistance in clinical settings.
- Point 4: For section 6.2, please provide more information about the mechanisms of ADC toxicity, as they are relevant to the strategies to overcome.
- Response 4: We appreciate the reviewer’s insightful comment. In response, we have revised Section 6.2 to incorporate additional mechanistic explanations of ADC-related toxicities, including ILD, hematologic toxicity, hepatotoxicity, and cardiotoxicity. Specifically, we describe immune-mediated inflammatory responses, off-target effects, systemic payload diffusion, Fc receptor-mediated endocytosis, and mitochondrial or oxidative stress as potential contributors to these adverse events.
- Point 5: The binding site barrier (BSB) has been proposed to be a challenge for ADCs in treating solid tumors. Please discuss this challenge.
- Response 5: We appreciate the reviewer’s valuable comment regarding the binding site barrier (BSB) as a critical challenge for ADC delivery in solid tumors. In response, we have expanded the manuscript to include a focused discussion on the BSB phenomenon, highlighting how ADCs preferentially bind to antigen-expressing tumor cells near the vasculature, thereby limiting deeper tissue penetration and causing heterogeneous intratumoral distribution. We further describe current strategies to overcome this barrier, including modifications of antibody affinity, reduction of ADC molecular size, the use of biparatopic or bispecific constructs, and application of site-specific conjugation techniques to enhance tumor penetration. These revisions improve the mechanistic insight into ADC delivery challenges in solid malignancies and are supported by appropriate references.
We tried our best to improve the manuscript and made some changes in the manuscript. These changes will not influence the content and framework of the paper. And here we did not list the changes but marked them in red in the revised paper.
We appreciate for Editors/Reviewers’ warm work earnestly and hope that the correction will meet with approval.
Once again, thank you very much for your comments and suggestions.

Round 2
Reviewer 1 Report (New Reviewer)
Comments and Suggestions for Authors
The authors responded appropriately to the reviewers' comments.
This manuscript is a resubmission of an earlier submission. The following is a list of the peer review reports and author responses from that submission.
Round 1
Reviewer 1 Report
Comments and Suggestions for Authors
The authors have an extensive list of citations but many of the generalizations made in the text and tables are in conflict with the current literature.
Author Response
- Point 1: The authors have an extensive list of citations but many of the generalizations made in the text and tables are in conflict with the current literature.
- Response 1: Thank you for your suggestion. After thorough re-evaluation, we recognize that certain statements in the main text and in Table 1 could be perceived as overly broad or not sufficiently aligned with recent findings in the literature.
In response to this concern, we have undertaken the following actions:
- We revised key generalizations throughout the manuscript to ensure they are accurately supported by the latest peer-reviewed data.
- In particular, in Table 1, we refined the comparative descriptions of DXd-based ADCs versus traditional chemotherapies to avoid oversimplification and to reflect mechanistic complexity and variability in toxicity profiles.
- We updated or replaced multiple references with more current and directly relevant studies to better support the discussed clinical implications and pharmacological properties.
- We believe that this revision significantly improves the scientific rigor and fidelity of our work to the current literature.

Reviewer 2 Report
Comments and Suggestions for Authors
The manuscript aims to provide an overview of DXd-based antibody-drug conjugates (ADCs), including mechanisms, clinical progress, and challenges of DXd-based ADCs, emphasising their role in reshaping modern oncology treatment paradigms. However, it does not successfully achieve these objectives. In several sections, it is unclear whether the authors are comparing DXd-based ADCs or ADCs in general to traditional chemotherapies. This lack of clarity creates confusion and undermines the manuscript’s purpose as a focused and informative review.
- Table 1 – The authors state tin table that traditional chemotherapies do not exhibit a bystander effect, in contrast to DXd-based ADCs. However, this comparison is not appropriate, as the concept of a bystander effect is not relevant to traditional chemotherapeutic agents. These agents are systemically distributed and act non-selectively on both cancerous and healthy proliferating cells. Therefore, the idea of a “neighboring cell” effect (central to the bystander mechanism in targeted therapies like ADCs) does not apply in the same mechanistic or conceptual context.
- Figure 1 – The structure labeled as “DXd” is inaccurate. The figure actually depicts the linker-payload structure (MC-GGFG-DXd), not the DXd payload alone.
- Trade names are inconsistently used—for example, “Enhertu” as trade name of U3-1402 (HER3-DXd) , but the corresponding name for Dato-DXd is omitted.
- The manuscript includes U3-1402 (HER3-DXd) in the section on FDA-approved ADCs, despite this agent not yet receiving regulatory approval.
- In the discussion of strategies to overcome challenges in DXd-based ADCs, the authors mention the use of non-cleavable linkers and cite T-DM- the authors earlier describe the linker used in DXd-based ADCs as stable and cleavable; if a non-cleavable linker were used instead, this would likely abrogate the bystander effect. The discussion lacks scientific depth and fails to explain the rationale or implications of the strategies mentioned.
- The authors state in abstract "challenges such as toxicity, resistance, and manufacturing scalability remain." However, the manuscript does not discuss manufacturing scalability challenges at all, which was one of main reason I accepted to review this paper
Author Response
- Point 1: The manuscript aims to provide an overview of DXd-based antibody-drug conjugates (ADCs), including mechanisms, clinical progress, and challenges of DXd-based ADCs, emphasising their role in reshaping modern oncology treatment paradigms. However, it does not successfully achieve these objectives. In several sections, it is unclear whether the authors are comparing DXd-based ADCs or ADCs in general to traditional chemotherapies. This lack of clarity creates confusion and undermines the manuscript’s purpose as a focused and informative review.
- Response 1: Thank you for your comment. We agree that certain sections of the manuscript may have lacked clarity regarding whether the comparisons were being made specifically between DXd-based ADCs and traditional chemotherapies, or ADCs in general. In response, we have revised the relevant sections—including Table 1—to explicitly state when comparisons pertain to DXd-based ADCs versus traditional chemotherapies. We also refined wording throughout the manuscript to consistently emphasize the unique features, mechanisms, and clinical relevance of DXd-based ADCs. These revisions aim to reinforce the manuscript’s original objective of providing a focused and informative review centered specifically on DXd-based ADCs.
- Point 2: Table 1 – The authors state tin table that traditional chemotherapies do not exhibit a bystander effect, in contrast to DXd-based ADCs. However, this comparison is not appropriate, as the concept of a bystander effect is not relevant to traditional chemotherapeutic agents. These agents are systemically distributed and act non-selectively on both cancerous and healthy proliferating cells. Therefore, the idea of a “neighboring cell” effect (central to the bystander mechanism in targeted therapies like ADCs) does not apply in the same mechanistic or conceptual context.
- Response 2: Thank you for your suggestion. Thank you for your detailed explanation of the mechanistic definition of the bystander effect. We agree that the concept of a bystander effect, as it pertains to ADCs, involves the local diffusion of membrane-permeable cytotoxic payloads to adjacent antigen-negative cells within the tumor microenvironment—a mechanism that is fundamentally distinct from the systemic, non-selective distribution of traditional chemotherapeutic agents.
In response to the reviewer’s comment, we have revised the language in Table 1 to remove the direct comparison and to avoid conceptual confusion. Instead of stating that traditional chemotherapies lack a bystander effect, we now clarify that the bystander effect is a unique feature of certain ADCs and not a relevant parameter in the context of conventional chemotherapy.
This change has been reflected in the revised Table 1.We thank the reviewer for pointing out this important conceptual distinction.
- Point 3: Figure 1 – The structure labeled as “DXd” is inaccurate. The figure actually depicts the linker-payload structure (MC-GGFG-DXd), not the DXd payload alone.
- Response 3: Thank you for your advice. We have revised Figure 1A of the manuscript based on your comments.
- Point 4: Trade names are inconsistently used—for example, “Enhertu” as trade name of U3-1402 (HER3-DXd), but the corresponding name for Dato-DXd is omitted.
- Response 4: Thank you for your comment. We fully agree that consistent trade names of ADCs is essential for clarity. In response, we have revised the manuscript to ensure uniform usage of both development codes and trade names for all relevant agents.
For instance, we now refer to Dato-DXd as “Dato-DXd (Dato-DXd, also known as datopotamab deruxtecan; trade name: DatrowayÒ)” where appropriate, similar to our treatment of EnhertuÒ (trastuzumab deruxtecan). If a trade name is not yet available or officially approved, we have clearly indicated this in the text.
- Point 5: The manuscript includes U3-1402 (HER3-DXd) in the section on FDA-approved ADCs, despite this agent not yet receiving regulatory approval.
- Response 5: Thank you for pointing out the mistake. In response to your comment, we have revised the section heading from “FDA-approved DXd-based ADCs” to “Clinical Progress and Regulatory Status of DXd-based ADCs” to more accurately reflect the approval status of the agents discussed. P-DXd (U3-1402, HER3-DXd) remains investigational, and this distinction is now clearly indicated in both the section title and content.
- Point 6: In the discussion of strategies to overcome challenges in DXd-based ADCs, the authors mention the use of non-cleavable linkers and cite T-DM- the authors earlier describe the linker used in DXd-based ADCs as stable and cleavable; if a non-cleavable linker were used instead, this would likely abrogate the bystander effect. The discussion lacks scientific depth and fails to explain the rationale or implications of the strategies mentioned.
- Response 6: Thank you for your insightful comment. We agree that the use of a non-cleavable linker, such as in T-DM1 (trastuzumab emtansine), contrasts mechanistically with the cleavable linker used in DXd-based ADCs, which enables the bystander effect through selective payload release in the tumor microenvironment. In response, we have revised Section 6.3 (Strategies to Overcome Challenges in DXd-based ADCs) to clarify the functional consequences of different linker types, particularly how non-cleavable linkers may reduce systemic exposure and off-target toxicity but also abrogate the bystander effect. We have also expanded the discussion to provide more mechanistic rationale for each strategy introduced.
- Point 7: The authors state in abstract "challenges such as toxicity, resistance, and manufacturing scalability remain." However, the manuscript does not discuss manufacturing scalability challenges at all, which was one of main reason I accepted to review this paper.
- Response 7: Thank you for this important comment. We acknowledge the omission of manufacturing scalability challenges in the original version. As this is a critical aspect for the clinical translation and commercialization of DXd-based ADCs, we have revised Section 6.3 (Strategies to Overcome Challenges in DXd-based ADCs) to include a dedicated discussion on the manufacturing challenges of ADCs, including batch consistency, DAR control, and bioprocess optimization strategies to improve scalability.

Reviewer 3 Report
Comments and Suggestions for Authors
This review provides a detailed summary of the development and clinical application of deruxtecan (DXd)-based antibody-drug conjugates (ADCs), focusing on their mechanisms of action, clinical efficacy, and ongoing challenges in precision oncology. The discussion effectively outlines the unique properties of DXd, including its role as a topoisomerase I inhibitor, its cleavable and stable linker design, and its high drug-to-antibody ratio, which contribute to its activity against HER2, HER3, and trophoblast cell surface antigen 2-expressing tumors such as breast and lung cancers. The review also covers the clinical progress of agents like trastuzumab DXd, patritumab DXd, and datopotamab DXd, highlighting their therapeutic potential. However, the manuscript would benefit from a more critical analysis of the limitations of DXd-based ADCs, particularly in terms of toxicity, resistance mechanisms, and the challenges associated with large-scale manufacturing. Additionally, while the review mentions these challenges, it could be strengthened by exploring potential strategies to address these issues, such as improvements in linker technology, alternative payloads, and strategies for better patient selection. Another aspect that could be improved is the scope of the review. Currently, the manuscript covers a broad spectrum of cancers, which makes it somewhat generalized. It may be more impactful to narrow the focus to a few well-characterized cancer types, such as breast, lung, and gastric cancers, where DXd-based ADCs have shown significant clinical promise. Furthermore, providing a summary of current and potential applications of DXd-based ADCs in specific cancer types would help to clarify their clinical utility and future research directions. This would not only make the review more focused but also more informative for researchers and clinicians looking to understand the targeted applications of these therapies.
Author Response
- Point 1: However, the manuscript would benefit from a more critical analysis of the limitations of DXd-based ADCs, particularly in terms of toxicity, resistance mechanisms, and the challenges associated with large-scale manufacturing.
- Response 1: Thank you for your suggestion. We agree that a more critical discussion of the limitations of DXd-based ADCs—including toxicity, resistance mechanisms, and manufacturing scalability—is essential to a balanced and informative review. In response, we revised Section 6.3 (Strategies to Overcome Challenges in DXd-based ADCs) to include additional content addressing these challenges in greater detail.
- Point 2: Additionally, while the review mentions these challenges, it could be strengthened by exploring potential strategies to address these issues, such as improvements in linker technology, alternative payloads, and strategies for better patient selection.
- Response 2: Thank you for this important suggestion. In response, we have expanded the latter part of Section 6.3 to discuss potential strategies aimed at overcoming the key limitations of DXd-based ADCs. Specifically, we incorporated content addressing the optimization of cleavable linker designs to improve enzymatic specificity and payload release, the development of alternative cytotoxic payloads beyond topoisomerase I inhibitors (e.g., DNA alkylating agents and tubulin inhibitors), and the use of biomarker-guided approaches for more precise patient selection. These additions provide a forward-looking perspective and contribute to a more comprehensive and translationally relevant review.
- Point 3: Another aspect that could be improved is the scope of the review. Currently, the manuscript covers a broad spectrum of cancers, which makes it somewhat generalized. It may be more impactful to narrow the focus to a few well-characterized cancer types, such as breast, lung, and gastric cancers, where DXd-based ADCs have shown significant clinical promise. Furthermore, providing a summary of current and potential applications of DXd-based ADCs in specific cancer types would help to clarify their clinical utility and future research directions. This would not only make the review more focused but also more informative for researchers and clinicians looking to understand the targeted applications of these therapies.
- Response 3: We sincerely appreciate the reviewer’s thoughtful suggestion regarding the scope of our review. As recommended, we have revised the manuscript to narrow its focus and enhance clinical relevance by emphasizing three major cancer types—breast, lung, and gastric cancers—in which DXd-based ADCs have demonstrated substantial therapeutic potential.
To address this point, we have newly added Section 5: Challenges and Clinical Applications of DXd-based ADCs in Breast, Lung, and Gastric Cancers, positioned between Sections 4 and 6. This section systematically summarizes the current clinical evidence for DXd-based ADCs in each of these cancer types and includes a comparative summary table outlining the clinical status, targets, and efficacy of each DXd-based ADC by cancer type.
Specifically:
- Subsection 5.1 highlights the expanding indications of T-DXd in HER2-positive and HER2-low breast cancers, and reviews the clinical development of P-DXd and Dato-DXd.
- Subsection 5.2 covers the efficacy of T-DXd in HER2-mutant NSCLC, as well as P-DXd and Dato-DXd in EGFR-mutant or TROP2-expressing lung cancers.
- Subsection 5.3 discusses the approval and clinical impact of T-DXd in HER2-positive gastric cancer, particularly referencing DESTINY-Gastric trials.
These revisions enhance the clarity and focus of our review, providing clinicians and researchers with a concise understanding of the most clinically validated applications of DXd-based ADCs and future directions for targeted therapeutic strategies.
